# A Low Iron Diet Protects from Steatohepatitis in a Mouse Model

**DOI:** 10.3390/nu11092172

**Published:** 2019-09-10

**Authors:** Lipika Salaye, Ielizaveta Bychkova, Sandy Sink, Alexander J. Kovalic, Manish S. Bharadwaj, Felipe Lorenzo, Shalini Jain, Alexandria V. Harrison, Ashley T. Davis, Katherine Turnbull, Nuwan T. Meegalla, Soh-hyun Lee, Robert Cooksey, George L. Donati, Kylie Kavanagh, Herbert L. Bonkovsky, Donald A. McClain

**Affiliations:** 1Center on Diabetes, Obesity and Metabolism, Department of Internal Medicine, Wake Forest Baptist Medical Center, Winston-Salem, NC 27101, USA; 2Department of Internal Medicine, University of Utah Medical Center, Salt Lake City, UT 84112, USA; 3Department of Internal Medicine, Wake Forest School of Medicine, Winston-Salem, NC 27157, USA; 4Sticht Center on Aging, Wake Forest School of Medicine, Winston-Salem, NC 27157, USA; 5Agilent Technologies, 121 Hartwell Ave, Lexington, MA 02421, USA; 6Department of Comparative Medicine, Wake Forest University, Winston-Salem, NC 27157, USA; 7VA Medical Center, Salt Lake City, UT 84148, USA; 8Department of Chemistry, Wake Forest University, Winston-Salem, NC 27109, USA; 9Department of Biomedicine, University of Tasmania, Hobart, Tasmania 7000, Australia; 10VA Medical Center, Salisbury, NC 28144, USA

**Keywords:** iron, NAFLD, RNA-seq, metabolic syndrome

## Abstract

High tissue iron levels are a risk factor for multiple chronic diseases including type 2 diabetes mellitus (T2DM) and non-alcoholic fatty liver disease (NAFLD). To investigate causal relationships and underlying mechanisms, we used an established NAFLD model—mice fed a high fat diet with supplemental fructose in the water (“fast food”, FF). Iron did not affect excess hepatic triglyceride accumulation in the mice on FF, and FF did not affect iron accumulation compared to normal chow. Mice on low iron are protected from worsening of markers for non-alcoholic steatohepatitis (NASH), including serum transaminases and fibrotic gene transcript levels. These occurred prior to the onset of significant insulin resistance or changes in adipokines. Transcriptome sequencing revealed the major effects of iron to be on signaling by the transforming growth factor beta (TGF-β) pathway, a known mechanistic factor in NASH. High iron increased fibrotic gene expression in vitro, demonstrating that the effect of dietary iron on NASH is direct. Conclusion: A lower tissue iron level prevents accelerated progression of NAFLD to NASH, suggesting a possible therapeutic strategy in humans with the disease.

## 1. Introduction

The prevalence of nonalcoholic fatty liver disease (NAFLD) is steadily increasing worldwide, mirroring increased rates of obesity and type 2 diabetes mellitus (T2DM) [1]. NAFLD represents a spectrum of conditions from simple steatosis to nonalcoholic steatohepatitis (NASH), cirrhosis, and hepatocellular carcinoma [2,3]. Hepatic iron has been studied as a biomarker of NAFLD. Pathologic iron overload, as seen in hereditary hemochromatosis, is associated with hepatic fibrosis, and hepatic iron levels correlate with the severity of NASH [4,5]. Furthermore, nearly one third of patients with NAFLD exhibit the “dysmetabolic iron overload syndrome,” as manifested by elevated serum ferritin with normal or mildly elevated transferrin saturation [6]. This suggests that in addition to pathologic iron overload, even the upper range of normal levels of iron, as might be seen in people with higher dietary intake, could also pose a danger to liver health as it does for T2DM risk [7,8,9]. Recently, investigators reported a dietary model of NASH using a high total and saturated fat diet with glucose and fructose supplementation of water (“fast food,” FF) that exhibits high fidelity to the human condition [10]. We have used this model to investigate the effects of iron on NAFLD progression by varying the dietary iron content in defined chow formulations. Levels of iron were chosen that ranged from low (4 ppm chow, still sufficient to maintain normal blood hemoglobin concentrations) [11], to low-normal (35 ppm, within standard chow content range), to high (2000 ppm, simulating the high end of human dietary iron intake). These iron-formulated diets were previously used by us to demonstrate the diabetogenic effects of iron [8]. Herein we report that low iron diets can delay and limit the progression of high-fat-diet-induced steatosis to NASH and fibrosis.

## 2. Materials and Methods

### 2.1. Animal Experimentation

All studies included in this publication were approved by IACUC protocol A17-017. Wild-type (WT) C57Bl6 male mice were fed standard chow until the start of the special diets. Normal chow (NC, 17%, 64%, and 19% kcal from fat, carbohydrates, and protein, respectively) and “fast food” (FF, 42%, 43%, and 15% kcal from fat (12% SFA, 0.2% cholesterol), carbohydrates, and protein, respectively) were formulated by Teklad/Envigo (Harlan Laboratories, Madison, WI, USA) to contain either: 4 mg, 35 mg, or 2 g iron/kg chow (low [LI], normal [NI], or high [HI] iron, respectively). Drinking water for the FF-fed mice contained 18.3 g/L glucose and 23.1 g/L fructose. Mice were started on the diets at 2–4 months of age, for a period of either 3–4 or 5–6 months before in vivo testing or tissue harvest. Individual ventilated caging (IVC) system (Allentown Inc., Allentown, PA, USA) with bed-o’cob bedding (The Andersons, Maumee, Ohio) was used to house the mice. Mice were single-housed to foster sedentary lifestyle during the dietary treatment and were given paper nesting material and paper bio-huts as enrichment. Food and water intake was monitored on a weekly basis. The mice on FF consumed more calories than on NC (13.5 ± 2.05 kcal/day on NC, 15.07 ± 3.11 kcal/day on FF, *p* < 0.0001), but there was no difference in caloric intake as a function of iron content of the two chows (*p* = 0.626). For intraperitoneal glucose tolerance testing, mice were fasted for 6 h. Mice were re-fasted overnight prior to euthanasia. All interventions were done during the light cycle. Fresh liver tissue was used for paraffin-embedded histology. NASH scores were calculated using rodent-specific histology criteria [12] (Appendix A), in a blinded semi-quantitative assessment by 2 trained readers, including a veterinarian scientist.

### 2.2. Study Approval

Liver biopsy samples were obtained intra-operatively from patients undergoing abdominal surgery (bariatric surgery, cholecystectomy, and hernia repair) in subjects according to protocols approved by the Wake Forest School of Medicine Institutional Review Board (IRB00007116 and IRB00049199). The study protocol conformed to the ethical guidelines of the 1975 Declaration of Helsinki as reflected in a priori approval by the IRB. Written informed consent was received from participants prior to inclusion in the study.

### 2.3. Cell Culture

TWNT-1 (JCRB1582 or Cellosaurus ExPASy # CVCL_J364), a human hepatic stellate cell line, was obtained from Japanese Collection of Research Bioresources (JCRB) Cell Bank/Sekisui XenoTech, LLC (US distributor) from Cambridge Kansas, USA. TWNT1 is an immortalized cell line established from LI90. Passage 19 was used for the data presented. HepG2, a hepatocyte cell line, was obtained from Cell Culture and Virus Vector Core (CVVCL) at Wake Forest Comprehensive Cancer Center (WFBCCC). Cells were tested for mycoplasma contamination prior to use in experiments. Cells were grown and maintained in 1X Dulbecco’s Modified Eagle Medium (DMEM) with 10% fetal bovine serum (FBS) and 1% Pen-Strep (100 IU/mL of Penicillin and 100 µg/mL of Streptomycin). Both cell types were maintained in log phase before plating onto 6-well plates at a count of 2.5 × 10^5^ cells per well. 2D co-culture was plated at 1.25 × 10^5^ cells of HepG2 and 1.25 × 10^5^ cells of TWNT1 to maintain the same cell counts. For the transwell experiments, Transwell^®^ Inserts (Costar Corning) with a 0.4 µm pore size and a 24 mm insert size were used. After 24 h of starvation in serum-free DMEM, cells were treated with iron or palmitate for an additional 24 h. Iron in the form of ferric ammonium citrate (Sigma Aldrich #F5879- Ammonium iron (III) citrate, St. Louis, MO, USA) was prepared as a 100 µM stock solution in water. Preparation of the free fatty acid, palmitic acid, was as described in Luo et al., 2012. Briefly, palmitic acid was conjugated to fatty acid free bovine serum albumin (BSA) by dissolving palmitic acid (Sigma Life Sciences #P0500, Steinheim, Germany) in 100% ethanol to make 200 mM solution. 10% fatty acid free-BSA was then mixed with this solution for 5 h to generate a 4 mM palmitate stock solution.

### 2.4. Commercial Assays

Serum measurements were carried out as follows: ferritin (Abcam Mouse Ferritin ELISA #ab157713, Abcam, Cambridge, MA, USA); insulin (CrystalChem Ultrasensitive Mouse Insulin ELISA #90080, Elk Grove Village, IL, USA); leptin (Quantikine Mouse/Rat Leptin ELISA #MOB00, Minneapolis, MN, USA); adiponectin (Abcam Mouse ELISA #ab108785, Cambridge, MA, USA); and a lipid panel with transaminases (Piccolo^®^ Lipid Panel Plus, Greisham, Germany). Hepatic measurements with colorimetric kits include the following: triglyceride (Cayman Chemical #10010303, Ann Arbor, MI, USA); protein carbonyl content (Abcam #ab126287, Cambridge, MA, USA); and citrate synthase activity (Cayman Chemical #701040, Ann Arbor, MI, USA). For citrate synthase activity measurement in human biopsy livers (Sigma Aldrich #CS0720, Poole, UK), an activity assay kit was used. 

### 2.5. MIP OES and LC-MS/MS

For hepatic iron content, approximately 30 mg of liver tissue was homogenized by sonication (Fisher Scientific Sonic Dismembrator F60, Fair Lawn, NJ, USA) in 300 µL of RIPA buffer (Roche Complete Mini #11836153001, Roche, UK). After increasing the volume to 600 µL with distilled water, 150 µL of 40% nitric acid (Fisher Scientific metal free grade, CAS #7697-37-2) was added. Samples were digested for 90 min at 95 °C and then centrifuged at 12,000 rpm for 20 min. Supernatants were diluted to a volume of 4 mL using 1% nitric acid and filtered through 0.45 µm nylon syringe filters (Fisher Scientific #09-719-008), prior to loading for run in the MIP OES (Microwave-Induced Plasma Optical Emission Spectrometer, Agilent 4200 MP-AES, Santa Clara, CA, USA). To measure redox state in the livers, reduced glutathione (GSH) and glutathione disulfide (GSSG) were quantified using LC-MS/MS (Liquid Chromatograph Mass Spectrometer, Shimadzu 8050, Japan).

### 2.6. RNA Sequencing

Total RNA was isolated from snap-frozen mouse livers, by a modified Trizol protocol, using the RNeasy mini kit (Qiagen #74106, Hilden, Germany). The quantity and quality of isolated RNA were determined by ultraviolet spectrophotometry and electrophoresis, respectively, on the Nanodrop 2000 (Thermo Fisher Scientific, Wilmington, DE, USA) and Agilent 2100 Bioanalyzer (Agilent Technologies, Santa Clara, CA, USA). cDNA libraries were prepared from QC-passed RNA using the Illumina^®^ TruSeq Stranded Total RNA with Ribo-Zero Gold Preparation kit (Illumina Inc., San Diego, CA, USA) and then sequenced on Illumina NextSeq 500 (San Diego, CA, USA) in two batches of 12 and 9 samples. An alignment of sequence reads was performed using STAR sequence alignment [13]; a gene-level summarization of reads (i.e., counts) was performed using FeatureCounts [14]. Differentially expressed genes (DEGs) were analyzed using DESeq2 [15] with gene counts as input. Significant DEGs were defined as *p* < 0.05 after adjustment for false discovery and average fold change between condition replicates of >2.0. DEGs were further analyzed using DAVIDv6.8 [16] and Ingenuity Pathway Analysis (IPA) [17]. 

### 2.7. Reverse Transcription and Real-time PCR Quantification

cDNA synthesis was performed using the SuperScript III First-Strand Synthesis System (Invitrogen, Thermo Fisher Scientific #18080051, Waltham, MA, USA) using oligo(dT) primers, an All-in-One First-Strand cDNA Synthesis Kit (GeneCopoeia #QP008), or a High Capacity cDNA Reverse Transcription Kit (Applied Biosystems #4368814, Waltham, MA, USA) using random as well as oligo dT primers. Power SYBR Green PCR Master Mix (Applied Biosystems #4367659) was used for qPCR.

### 2.8. Seahorse Mito Stress Test

For the respirometric assay, mitochondria were isolated from fresh tissue using a modified version of the protocol described by Bharadwaj et al. [18]. In brief, approximately 100 mg of liver tissue was homogenized using a glass Dounce homogenizer in Chappel–Perry buffers I and II. After centrifugation at 600× *g* for 10 min at 4 °C, the supernatant was passed through a wetted cheese cloth to remove the non-mitochondrial fractions. Mitochondria were then pelleted by ultra-centrifugation at 10,000× *g* for 10 min at 4 °C. Five or 10 µg of mitochondrial protein was loaded into the Seahorse Bioscience XF^e^24 extracellular flux assay plate in a mannitol-sucrose-rich mitochondrial assay solution (MAS) buffer with succinate/rotenone (complex II conditions) with freshly prepared adenosine diphosphate (ADP), oligomycin, carbonyl cyanide-4 (trifluoromethoxy) phenylhydrazone (FCCP), and antimycin inhibitor drugs.

### 2.9. Statistics

Statistical analyses were performed using GraphPad Prism V7. Grouped analyses were done using the ordinary two-way ANOVA with post hoc Tukey’s multiple comparisons test. Error bars in the figures represent ± SEM. Tabular statistical results of RNA Seq analyses were auto-generated by DAVID or IPA Core analysis. Clinical data were analyzed by the linear regression model.

## 3. Results

### 3.1. Metabolic Phenotyping of Mice on Diet for 3–4 Months

The effects of FF began to be manifest 3–4 months after initiation of the diets. At this time, mice on FF were heavier compared to those on normal chow (NC) (Figure 1A). Although fasting glucose values did not differ (Figure 1B), the area under the glucose curve (AUC) during intraperitoneal glucose tolerance testing was higher in the FF group overall, though only the high iron (HI) FF group differed significantly from the corresponding NC group (Figure 1C). Beta cell function and insulin resistance as estimated by the homeostasis model assessment (HOMA-B/IR) were not statistically different among the groups (Figure 1D,E). Serum cholesterol (Figure 1F) and serum triglycerides (TG) (Figure 1G) were higher in the FF diet groups. Thus, after 3–4 months on the diets, the effects of FF on weight and glycemia were modest, and insulin resistance was not yet established. After 5–6 months on the diets, the FF groups demonstrated further body weight gain (Figure 2A), higher fasting glucose values (Figure 2B), and an increased AUC (Figure 2C). Higher HOMA-B in the FF group reflected hyperinsulinemia, which diminished at higher iron (Figure 2D). HOMA-IR values were also higher in the FF group, and this too diminished in the HIFF group (Figure 2E). Similar to 3–4 months, the FF group exhibited hypercholesterolemia, although serum TG remained largely unaffected by either diet or iron (Figure 2F,G). 

### 3.2. Iron Indices Are Not Altered by the FF Diet, and Hepatic Fat Content Is Not Affected by Dietary Iron

In order to analyze the effects of dietary iron on NAFLD, we first needed to determine if the FF diet interfered with normal iron homeostasis or tissue iron accumulation, and conversely, if dietary iron affected liver fat accumulation. Serum ferritin, a marker for tissue iron, mirrored dietary iron content in both the FF and NC groups, at 3–4 months (Figure 3A) and 5–6 months (Figure 3B). The same was true for hepatic iron content (Figure 3C,D). Gene expression of hepcidin (Hamp), a major hormonal regulator of iron homeostasis, also increased significantly with dietary iron in both diets (Figure 3E), although at 5–6 months hepcidin levels were higher in the NC group compared to FF (Figure 3F). Transcript levels of bone morphogenetic protein 6 (Bmp6), an endogenous regulator of hepcidin expression, increased with iron, although the level of expression and effect of iron was larger in the FF group at 5–6 months (Figure 3G,H). Thus, tissue iron levels and serum ferritin in the mice on the various iron diets were largely unaffected by whether the diets were FF or NC, although there were some effects of diet on Hamp and Bmp6. Regarding iron’s effect on fat accumulation, hepatic TG content was higher in FF groups at 3–4 months on the diet (Figure 3I), and this effect became more significant after 5–6 months (Figure 3J). Dietary iron, however, did not affect hepatic TG at either time (Figure 3I,J). Histologic analysis of the liver tissue stained for neutral lipid with Oil Red O showed a similar pattern of results (Appendix A). Thus, hepatic steatosis was affected by FF, but not by dietary iron. 

### 3.3. Iron Restriction Limits Progression of NAFLD to NASH

Given the facts that FF did not affect tissue iron accumulation, and that dietary iron content did not affect steatosis, we could isolate these variables in the progression of liver injury. We first examined serum alanine aminotransferase (ALT), a marker for liver injury. At 3–4 months on the diet, serum ALT levels were within the normal range, although the FF groups trended higher compared to NC (Figure 4A). By 5–6 months on the diet, ALT levels were significantly higher in the FF group. However, ALT in the low iron (LI) FF group was significantly lower than the two higher iron FF groups (Figure 4B). These studies employed male mice. Similar results are seen in females, although the degree of ALT elevations was lower, consistent with the gender-specific pattern of severity seen in humans (Appendix A).

We next examined the inflammatory mediators tumor necrosis factor alpha (TNF-α) and transforming growth factor beta (TGF-β). At 3–4 months on the diets, Tnfa transcript levels increased with FF, but not significantly with iron (Figure 4C). At 5–6 months, Tnfa mRNA was lower on the LI and normal iron (NI) diets compared to the higher iron diets for both NC and FF groups, though this was significant only in the NC diet (Figure 4D). Tgfb mRNA increased dramatically from 3–4 months to 5–6 months on the diets, and was higher in the FF group but not significantly affected by iron (Figure 4E,F). The transcript expression of collagen1alpha1 (Col1a1), an extracellular matrix protein involved in hepatic fibrogenesis, increased several-fold from 3–4 months to 5–6 months on the FF NI and HI diets, but was strikingly lower in the LIFF group (Figure 4G,H). 

### 3.4. Iron Transcriptionally Regulates the TGF-Β Signaling Pathway

Transcriptome profiling by RNA sequencing revealed that the expression of multiple genes is altered by dietary iron (Figure 5A,B). To analyze differential gene expression, we chose the LIFF and HIFF groups that differed in terms of hepatic inflammation and fibrosis. We generated a dataset of DEGs that exhibited at least two fold changes in expression, significant at *p* < 0.05 after adjustment for false discovery. Pathways with false discovery rates >10 were not considered, even with low p values. Pathway enrichment analysis using DAVIDv6.8 identified a single Kyoto Encyclopedia of Genes and Genomes (KEGG) pathway (mmu04350: TGF-β signaling pathway) as significantly enriched, at both 3–4 months and 5–6 months (Appendix A). 

In the TGF-β signaling pathway, DEGs from our dataset included inhibitors of differentiation/DNA binding (Id1, Id2, Id3, and Id4) and non-canonical Smad (Drosophila, mothers against decapentaplegic) genes (Smad7, Smad9). A schematic visualization was generated using IPA with the criteria for core analysis as *p* < 0.05 and log2FC > 0.8 or < −0.8 (Appendix A). IPA analysis showed the TGF-β signaling pathway as the third most significant canonical pathway with a positive activation z score of 1.00 for the low iron group, meaning the pathway is inhibited in the HIFF group, both at 3–4 months (Figure 5E) and 5–6 months. Of the 96 genes in the canonical TGF-β signaling pathway, 44.8% are downregulated, 45.8% are upregulated, and 9.4% had no overlap in our dataset (Figure 5F). Gene ontology (GO) analysis further showed that genes most significantly regulated by iron are those related to the extracellular space including Col1a1 and Col1a2 (Appendix A). IPA also identified potential upstream regulators enriched in the dataset, predominantly inflammatory cytokines at 3–4 months and growth factors at 5–6 months (Appendix A). The analysis also revealed a tumorigenic potential in the HIFF transcriptional profile as early as 3–4 months (Appendix A).

We generated a second DEG dataset to assess the effect of FF on the hepatic transcriptome (NINC vs NIFF), which included important regulators of lipid metabolism including squalene epoxidase (Sqle), lipoprotein lipase (Lpl), and fatty acid binding protein 4 (Fabp4). Functional annotation corroborated canonical pathways enriched for lipid metabolism and predicted biofunctions related to organismal injury, cancer, inflammatory response, and immune cell trafficking (Appendix A). 

A comparative analysis between the two DEG datasets identified a subset of 10 genes that were affected by both iron and FF diet (Figure 5D and Appendix A). These include the transferrin receptor (Tfrc), indicating crosstalk between iron and lipid metabolism, and Serpine1, implicated in fibrosis, suggesting that fibrogenesis is regulated by both steatosis and iron.

### 3.5. Hepatocyte–Stellate Cell Interaction is Necessary for Iron- and Fat-Induced Fibrogenesis

To test our hypothesis that iron directly affects the transcriptional regulation of fibrogenic pathways, we conducted in vitro experiments. The hepatoma-derived cell line HepG2 and the hepatic stellate cell (HSC) line TWNT1 were used to study the effect of iron on fibrotic gene expression. Palmitate, a saturated fatty acid, was our surrogate for the FF diet, and ferric ammonium citrate (FAC) was the source of iron. Treatment of the TWNT1 cells with palmitate plus iron showed an increase in the gene expression of α-smooth muscle actin (ACTA2 gene), a marker for activated HSCs. This effect was greater when the TWNT1 cells were exposed to conditioned medium from iron- and palmitate-treated HepG2, in a transwell. This effect amplified when TWNT1 and HepG2 cells were co-cultured in the same well (Figure 6). 

### 3.6. Oxidative Stress and Other Previously Identified Mediators Are Not Predominant Early Drivers of NASH in This Model

Oxidative stress, as measured by the GSH/GSSG ratio (Appendix A), and protein carbonyl content (Appendix A) were not significantly different among the groups. Serum adipokines, leptin (Appendix A), and adiponectin (Appendix A) were also not significantly different, although there was an iron-dosage-dependent trend in the serum levels. 

### 3.7. Iron Does Not Affect Mitochondrial Function Prior to the Initiation of Fibrosis

In the fibrosis of other organs, including the liver, diminished mitochondrial function has been proposed to play a role in fibrogenesis. We therefore examined the functional capacity of partially purified mitochondria from livers of mice who had been on the diets for 5–6 months. The HI diet had no significant effect on basal or maximal oxygen consumption rates, proton leak, or spare respiratory capacity in mitochondria from mice on either NC or FF diets (Appendix A). Likewise, no effects were seen on ATP production or coupling efficiency (Appendix A). There was a trend for oxygen consumption rates and proton leak to be lower in the mice on low iron (*p* = 0.49 and *p* = 0.19 respectively). There was no loss of mitochondrial mass either, as indicated by citrate synthase activities of liver extracts from these mice (Appendix A) and by an analysis of liver biopsy specimens from individuals undergoing abdominal surgery (Appendix A).

## 4. Discussion

We have shown that limiting dietary iron limits or delays NASH progression in a dietary (“fast food”) mouse model of NASH. Low dietary iron protects mice on the FF diet from developing elevations in the cardinal clinical marker of NASH and ALT and from transcriptional markers of inflammation and fibrogenesis. These effects are seen early and in some cases prior to the onset of full-blown metabolic disease. Importantly for these analyses, over the time frame of this study, the FF diet did not affect systemic iron levels, nor did dietary iron significantly alter hepatic steatosis, allowing the effects of each variable to be analyzed independently of the other. We believe the levels of dietary iron recapitulate the broad range of human tissue iron levels seen at the lower and upper limits of normal, e.g., in vegans and in heavy consumers of red meat. For example, the ~10-fold range in hepatic iron levels in the general human population [19] is comparable to the range seen in these murine models [20].

Many animal models of NASH are based on nutrient deficiency, acute drug-induced effects, or genetic manipulations. The dietary model that we used, by contrast, displays high fidelity to human condition, in being slower in onset and driven by dietary factors [10]. On a molecular pathway level, the gene expression patterns in this model are also similar to those observed in humans [21]. More recent data also indicate that overt fibrosis and hepatocellular carcinoma (HCC) develop in mice on the diet for extended periods [22]. Murine models of pathologic iron overload (e.g., hereditary hemochromatosis) also reflect the corresponding human conditions to a great degree [23], although one weakness has been their failure to consistently develop NASH and cirrhosis [24,25]. We have recently found that NASH does develop in Hfe-/- mice on the FF diet even at lower levels of dietary iron (manuscript in preparation). It should be noted that that C57Bl6 mice typically take 7 months or longer to develop visible fibrosis on FF [26]. Consistent with this, fibrosis was not yet evident by histologic staining in our mice at 5–6 months (Appendix A), nor did qualitative NASH scoring of liver histology yield significant differential scores on liver histology (Appendix A). On an extended period of diet, however, we saw significant fibrosis visible in histology (Appendix A). 

Whole transcriptome RNA sequencing was performed to further analyze the effect of dietary iron on NASH pathogenesis. The transcriptional pathways most significantly altered by higher levels of dietary iron, within the FF-diet groups, are the mouse embryonic stem cell pluripotency, BMP signaling, and TGF-β signaling pathways. Induction of stem-cell pluripotent genes is indicative of cellular reprogramming to a dedifferentiated state that occurs during HCC development [27]. BMP signaling has a complex role in regulating iron homeostasis, liver fibrosis, and liver regeneration [28]. TGF-β is known to induce transdifferentiation and production of collagen and extracellular matrix proteins in hepatic stellate cells, and deletion of the TGF-β receptor in hepatocytes has been shown to alleviate NASH [29]. Increased expression of TGFB in NASH patients has also been reported [30,31]. Here, we show that iron and the FF diet independently drive TGF-β. The elevations of Tgfb in the mice on HINC (Figure 4F) suggest that TGF-β may be necessary but not sufficient for NASH development in this model. Other iron-regulated genes included those belonging to the inhibitor of differentiation/DNA binding (Id) and mothers against the decapentaplegic (Smad) family of genes. Id1-4 genes, previously implicated as mediators of TGF-β signaling and metabolic reprogramming in cancer models, including HCC, were increased in high iron group, as compared to low iron FF group [32,33]. Many metabolic hallmarks of cancer, notably, increased dependence on non-oxidative metabolism, are mirrored by metabolic effects of iron [11]. ID1 and 2 are regulated by hypoxia-inducible factor 1 (HIF1A), BMP6, and MAPK/ERK pathways, all of which are also modulated by iron [11,34]. From the SMAD family of genes, SMAD7 and SMAD9 were increased in the high iron FF group. SMAD7, a negative regulator of TGF-β signaling [35] and hepcidin expression [36], when deleted from hepatic tissue, causes spontaneous liver injury and aggravates alcoholic liver damage [37]. The regulation of TGF-β signaling by iron and the lack of induction of the genes in that pathway in the low iron diet mice suggest that modulation of TGF-β signaling is the mechanistic basis for the protection afforded by low iron. 

After 3–4 months on FF, hepatic steatosis was evident although indices of glucose tolerance and insulin sensitivity were minimally affected. Inflammatory markers of NASH including Tnfa [38], however, were significantly increased and serum ALT was beginning to rise. By 5–6 months, these differences were more fully established, and a transcriptional marker of fibrosis, Col1a1, was also elevated with FF. On the FF diet, LI protects from elevations in ALT, Tnfa, and Col1a1. The least insulin resistant FF mice, those on HI, have the highest elevations of serum ALT and Col1a1, demonstrating that insulin resistance is not well-correlated with the evolution of NASH in this model, and that other drivers including iron may be more important. 

We also investigated the role of iron in affecting other proposed mediators of NASH progression such as oxidant stress [39]. We were unable, for example, to demonstrate significant effects of iron on the ratios of reduced to oxidized glutathione. The serum adipokines, leptin, and adiponectin, are also known to play a role in NAFLD and HCC [40], and we have previously reported that both are regulated by iron [18,41,42]. However, at the early time points in this analysis, during which the pathogenic processes of inflammation and fibrosis were being initiated, we did not see significant alterations in either adipokine. Another proposed mediator of the NAFLD to NASH transition is mitochondrial dysfunction [43]. The early evolution of NASH seen in this study, however, precedes mitochondrial dysfunction, and the early protection from NASH progression afforded by the low iron diet is not related to preservation of mitochondrial function; in fact, the LI diet groups tended to have lower parameters of respiration. There was also no significant effect of iron on a surrogate marker of mitochondrial mass, citrate synthase activity, either in mice or in the clinical liver biopsies of patients undergoing abdominal surgery.

The above observations support the hypothesis that progression of NAFLD to NASH is a multiple-hit process involving factors including steatosis, insulin resistance, cytokines, tissue iron, and others, and that different combinations of factors may be sufficient for pathogenesis depending on their degree of abnormality [44,45]. The pathogenesis of NASH involves the interaction of multiple cell types. Although the hepatocyte is the most directly affected by and responsible for steatosis, the primary source of ECM proteins that initiate fibrogenesis in the liver are hepatic stellate cells [46,47]. TGFβ is a potent activator of hepatic stellate cells, and since we hypothesized that iron acts through the TGFβ signaling pathway to promote hepatic fibrogenesis in NASH, we tested our hypothesis in a two-cell in vitro model. Stellate cell activation was indeed enhanced when co-cultured with a hepatocyte cell line and treated with palmitate and iron. Thus, we propose a mechanistic model of iron-associated NASH pathogenesis, in which transcriptional changes mediated by the TGFβ signaling pathway promote hepatic fibrogenesis and a predisposition to HCC.

The association of the dysmetabolic iron overload syndrome with NASH progression supports iron reduction as a potential treatment strategy for NASH. Some clinical trials have shown that iron reduction by phlebotomy or chelation therapy can ameliorate the serum markers and/or histopathological features of NASH [47,48,49,50], although others have failed to show benefit [51,52], in some cases possibly related to the studies not following subjects for a sufficiently long time to detect beneficial effects. Our data support a potential role for iron reduction at least for prevention, but more study is needed, particularly clinical trials of sufficient size and duration to detect those potential benefits. This is especially true given that fact that iron reduction by phlebotomy is a safe and acceptable treatment of hereditary hemochromatosis and would be available to patients without further regulatory approvals.

## 5. Conclusions

We have shown that low tissue iron protects from the initiation and progression of NASH in a dietary mouse model of the disease, and that many of the effects of iron may be exerted through TGF-β-dependent pathways. The key genes involved in this iron-altered pathogenesis could serve as diagnostic or prognostic tools. Of more potential importance is the possibility that abnormal iron levels, arising not only from hereditary causes or excess transfusions but also from dietary excess, can be manipulated to alter the course of the disease.

## Figures and Tables

**Figure 1 nutrients-11-02172-f001:**
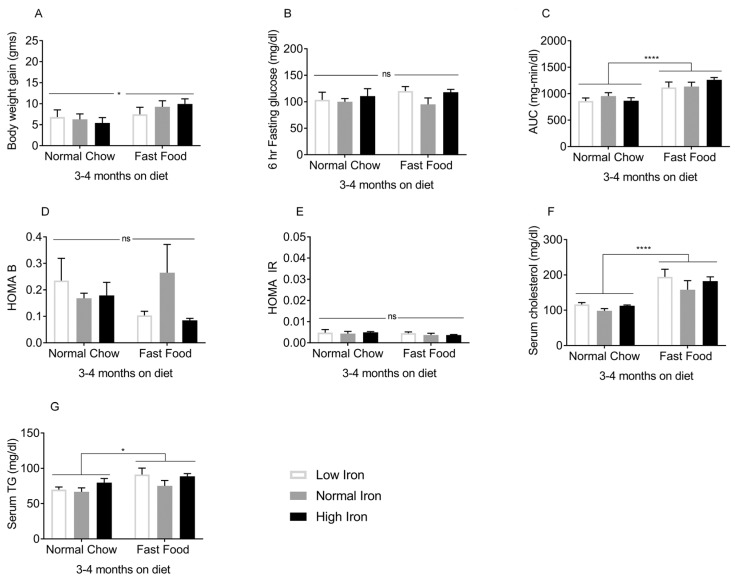
Metabolic phenotypes after 3–4 months on diet: The following were measured in mice on the FF/NC diets for 3–4 months (*n* = 3–5 per group). (**A**) Body weight. (**B**) Blood glucose after 6 h fasting. (**C**) Glucose tolerance (area under the glucose curve [AUC] after intra-peritoneal glucose tolerance testing). (**D**) Homeostasis model assessment of β-cell function (HOMA B). (**E**) Homeostasis model assessment of insulin resistance (HOMA IR). (**F**) Serum cholesterol and (**G**) Fasting serum triglycerides (TG). Shown are means ± SE. *p* values were calculated by two-way ANOVA with post hoc Tukey’s multiple comparisons test. * 0.01 ≤ *p* ≤ 0.05; **** *p* < 0.0001, ns: not significant, i.e., *p* ≥ 0.05.

**Figure 2 nutrients-11-02172-f002:**
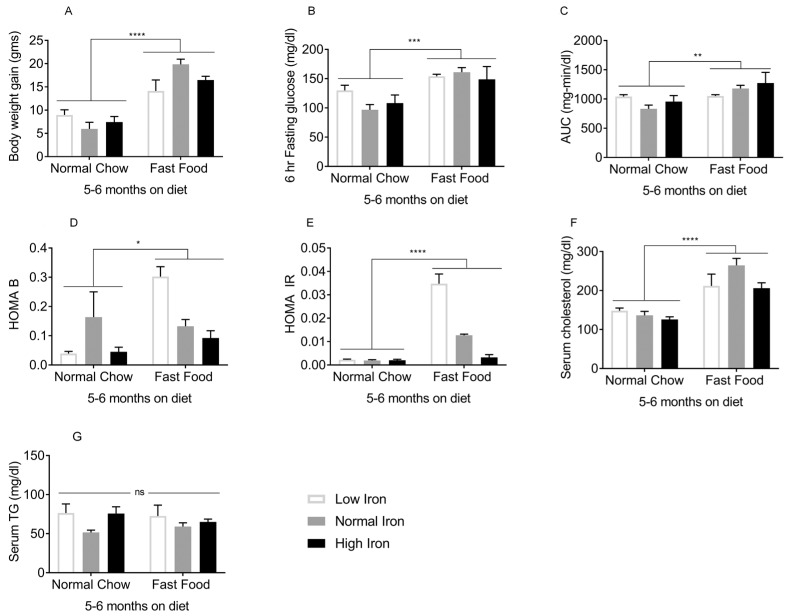
Metabolic phenotypes after 5–6 months on diet: The following were measured in mice on the FF/NC diets for 5–6 months (*n* = 8–10 per group). (**A**) Body weight. (**B**) Blood glucose after 6h fasting. (**C**) Glucose tolerance (area under the glucose curve [AUC] after intra-peritoneal glucose tolerance testing). (**D**) Homeostasis model assessment of β-cell function (HOMA B). (**E**) Homeostasis model assessment of insulin resistance (HOMA IR). (**F**) Serum cholesterol and (**G**) Fasting serum triglycerides (TG). Shown are means ± SE. *p* values were calculated by two-way ANOVA with post hoc Tukey’s multiple comparisons test. * 0.01 ≤ *p* ≤ 0.05; ** 0.001 ≤ *p* ≤ 0.01; *** 0.0001 ≤ *p* ≤ 0.001; **** *p* < 0.0001. ns: not significant.

**Figure 3 nutrients-11-02172-f003:**
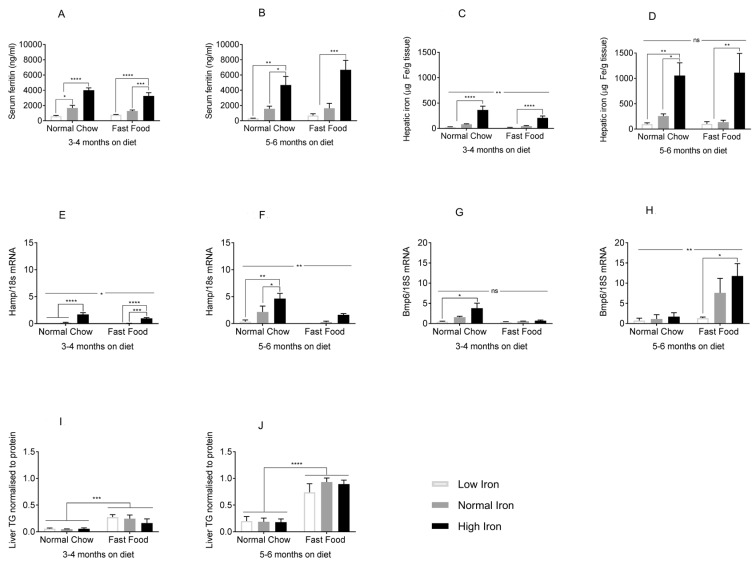
Effects of the FF diet on iron parameters and of dietary iron on hepatic lipid accumulation. (**A**,**B**) Serum ferritin at 3–4 and 5–6 months (*n* = 3–5 per group). (**C**,**D**) Hepatic iron content at 3–4 and 5–6 months (*n* = 3–6 per group). (**E**,**F**) Hepcidin mRNA (Hamp) normalized to 18s rRNA at 3–4 and 5–6 months (*n* = 3–5 per group). (**G**,**H**) Bone morphogenetic protein 6 (Bmp6) mRNA at 3–4 and 5–6 months (n = 3–5 per group). (**I**,**J**) Liver triglyceride content (TG) at 3–4 and 5–6 months (*n* = 3–10 per group). Shown are means ± SE. *p* values were calculated by two-way ANOVA with post hoc Tukey’s multiple comparisons test. ns: not significant, i.e., *p* ≥ 0.05; * 0.01 ≤ *p* ≤ 0.05; ** 0.001 ≤ *p* ≤ 0.01; *** 0.0001 ≤ *p* ≤ 0.001; **** *p* < 0.0001.

**Figure 4 nutrients-11-02172-f004:**
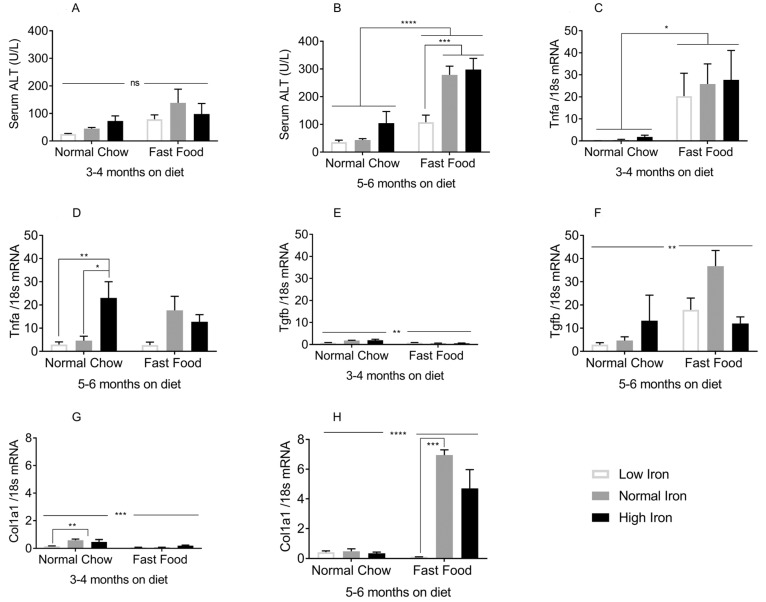
Effects of dietary iron and FF on NASH biomarkers: (**A**,**B**) Serum ALT at 3–4 and 5–6 months (*n* = 3–11 per group). (**C**,**D**) Tumor necrosis factor alpha (Tnfa) mRNA at 3–4 and 5–6 months (*n* = 3–5 per group). (**E**,**F**) Transforming growth factor beta (Tgfb) mRNA at 3–4 and 5–6 months (*n* = 3–5 per group). (**G**,**H**) of collagen1alpha1 (Col1a1) mRNA at 3–4 and 5–6 months (*n* = 3–5 per group). All genes were normalized to 18sRNA. Shown are means ± SE. p values were calculated with ordinary two-way ANOVA with post hoc Tukey’s multiple comparisons test. ns: not significant, i.e., *p* ≥ 0.05; * 0.01 ≤ *p* ≤ 0.05; ** 0.001 ≤ *p* ≤ 0.01; *** 0.0001 ≤ *p* ≤ 0.001; **** *p* < 0.0001.

**Figure 5 nutrients-11-02172-f005:**
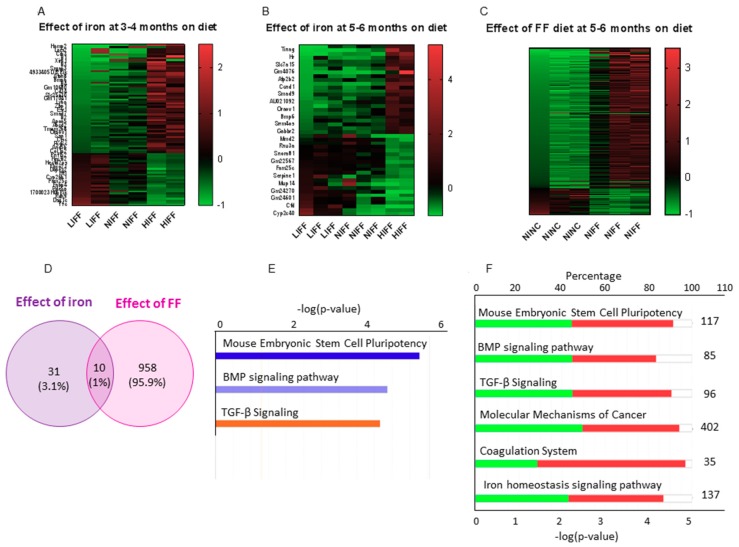
Differential gene expression in the hepatic transcriptome induced by iron and FF: (**A**–**C**) Heat maps of DEG using cutoff log2FC >1 or <−1, *p* < 0.01 on DeSEq2 data. Each row represents a DEG and each column is a sample from an individual mouse fed the diet as indicated by the column label. (**A**) Effect of iron in the FF-fed mice at 3–4 months. First two columns: low iron (LI) FF; next two columns: normal iron (NI) FF; last two columns: high iron (HI) FF. (**B**) Effect of iron in the FF-fed mice at 5–6 months; three samples each for LIFF and NIFF and two samples for HIFF. (**C**) Effect of FF compared to normal chow (NC). First three columns: LINC; next three columns: LIFF. (**D**) Venn diagram of DEGs affected by iron and FF, showing only 1% of genes altered by both iron and FF. (**E**) Bar chart of top 3 canonical pathways enriched by IPA core analysis. (**F**) Stacked bar chart of the number of genes upregulated and downregulated in the top 6 canonical pathways enriched by IPA core analysis.

**Figure 6 nutrients-11-02172-f006:**
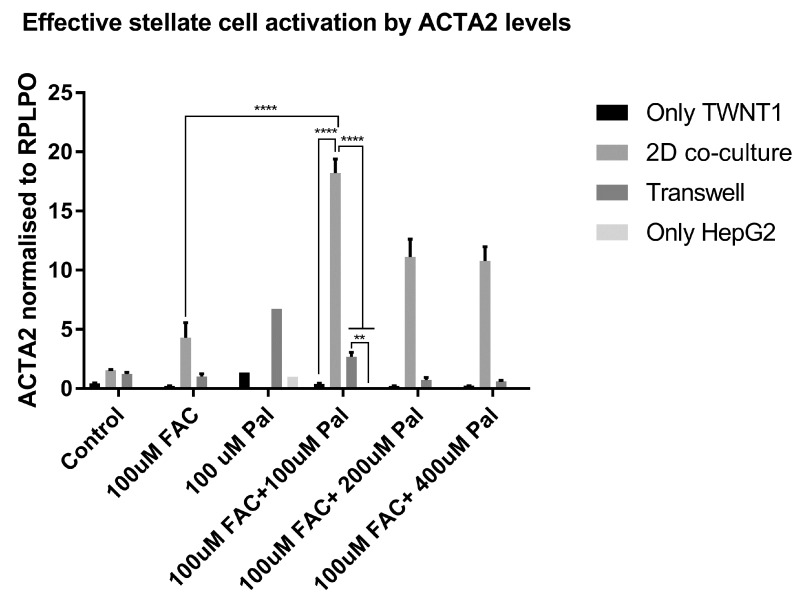
Hepatocyte-stellate cell interaction is necessary for iron- and fat-induced fibrogenesis: mRNA expression of smooth muscle actin (ACTA2), upon treatment with iron (FAC) and palmitate (Pal). *n* = 3 biological replicates. Error bars are shown as means ± SE. *p* values were calculated by two-way ANOVA with post hoc Tukey’s multiple comparisons test. ** 0.001 ≤ *p* ≤ 0.01; **** *p* < 0.0001.

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
