# Peer review of "A Low Iron Diet Protects from Steatohepatitis in a Mouse Model"

_nutrients, 2019, doi:10.3390/nu11092172_

Round 1

Reviewer 1 Report

The authors investigated relationships between iron intake and western diet (e.g., high fat and high fructose) in NAFLD condition in mice. Based on their findings, the authors concluded low tissue iron level alleviated progression of NAFLD to NASH possibly via TGF-beta pathway. This study provides some interesting data with interesting study model and disease condition but requires some major revisions and justifications as detailed below.

Major comments

Food intake is missing. In fact, calorie intake data might be more informative as these diets are NOT isocaloric diets. Should provide justification(s) how palmitate can serve a surrogate for FF diet of your animal study especially without having fructose in the media. Lacks on discussion as to the other top 2 most significantly impacted canonical pathways suggested by the IPA. Require more validation data from in vitro treatment conditions especially as to TGF-Beta signaling pathway. Especially, line 382 through 384 might be overstated without having more validation data in relation to TGF-beta signaling.

Minor comments

So many typos and inconsistency throughout the manuscript including, but NOT limited to:

There should be spaces between units and numbers throughout the manuscript (e.g., Line 95 or Line 98). FFA should be defined (line 98). Line 137: reference number 105? Check the degree unit on line 139 and 141. Line 213, AST and ALT were already used in the method and now is abbreviated. Line 241, DEGs was already defined in the method. Line 249, IPA was already defined in the method. Why supplementary figures are present in the main text? Line 375, HCC was already defined in the text.

I can’t not even list all typos in this review letter so it is strongly recommended for the authors to carefully review the manuscript and correct them to present professionally.

Reviewer 2 Report

The manuscript investigates the role of role of iron in development of non-alcoholic steatohepatitis in mice model fed with high fat diet with supplemental fructose in water. In addition human hepatic stellate cell and HepG2 cell line have been used to test fibrogenic process of liver.

The manuscript is well written and presented. There is one major concern which is the ratio of reduced glutathione vs oxidized glutathione (GSH/GSS). This ratio in various body organs ranges from 40-60. In the current study, this ratio is shown around 3000 (arbitrary unit).

The liver GSH level could be 4-6 µmol/g liver while GSSG values could be between 0.09-0.11 µmol/g tissue. A ration as shown above seems not correct.

It is difficult to reconcile the data in “Supp fig 6A” for hepatic GSH/GSSG ratio in two groups, i.e., low iron and High iron. The bar diagram shows that ratio of GSH/GSSG is approximately around 3000. The data means that there are almost no oxidized glutathione (GSSG). It should be checked carefully.

Correct the sentence

Line 199-201: Histologic analysis the liver tissue stained for neutral lipid with Oil RedO showed a similar pattern of results (Supplementary Figure 1). Thus, hepatic steatosis was affected by FF, but not by dietary iron.

Round 2

Reviewer 1 Report

The manuscript was improved after the revision. Although there might be some minor typos those can be checked again through the publication process.